# OPTIMAL STEPSIZE FOR DIFFUSION SAMPLING

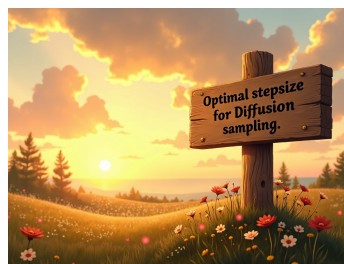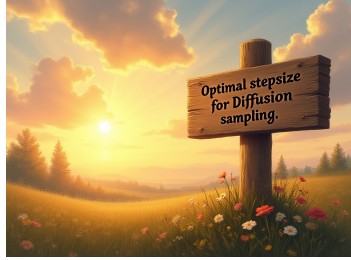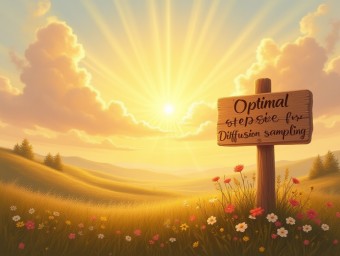

Figure 1: Flux sampling results using different stepsize schedules. Left: Original sampling result using 100 steps. Middle: Optimal stepsize sampling result within 10 steps. Right: Naively reducing sampling steps to 10.

## ABSTRACT

Diffusion models achieve remarkable generation quality but suffer from computational intensive sampling due to suboptimal step discretization. While existing works focus on optimizing denoising directions, we address the principled design of stepsize schedules. This paper proposes Optimal Stepsize Distillation, a dynamic programming framework that extracts theoretically optimal schedules by distilling knowledge from reference trajectories. By reformulating stepsize optimization as recursive error minimization, our method guarantees global discretization bounds through optimal substructure exploitation. Crucially, the distilled schedules demonstrate strong robustness across architectures, ODE solvers, and noise schedules. Experiments show **10×** accelerated text-to-image generation while preserving **99.4%** performance on GenEval.

## 1 INTRODUCTION

Score-based generative models Song et al. (2020b); Song & Ermon (2020) have emerged as a groundbreaking paradigm in generative modeling, achieving state-of-the-art results across diverse domains. Unlike traditional approaches that directly model complex data distributions, these methods decompose the target distribution into a sequence of conditional distributions structured as a Markov chain. While this decomposition enables tractable training through iterative noise prediction, it inherently necessitates computational expensive sampling procedures.

From a theoretical perspective, diffusion models approximate the data generation process as a continuous transformation from noise to structured data, parameterized by an infinite series of conditional distributions mapping. But in practice, computational constraints require discretizing this continuous trajectory into a finite sequence of steps, introducing a critical trade-off between sampling efficiency and approximation fidelity.

The discretization includes two key factors: the update direction (determined by the score function) and the stepsize (controlling the progression towards the target distribution), as shown in Figure 2. While significant attention has been devoted to optimizing update directions, including higher-order solvers Lu et al. (2022a;b) and corrected network predictions Zhao et al. (2024), the principled design of stepsize remains underexplored. Existing methods predominantly rely on heuristic schedules, such as the uniform timesteps in DDIM Song et al. (2020a) or the empirically designed schedule in EDM Karras et al. (2022), lacking theoretical guarantees of optimality under constrained steps.

In this work, we propose a principled framework for searching the **O**ptimal **S**tepsize in diffusion **S**ampling (**OSS**), compatible with arbitrary direction strategies. Specifically, we regard the stepsize

searching problem as a knowledge distillation problem, where a "student" sampling process with limited steps aims to approximate the results of a "teacher" sampling process with abundant steps. Crucially, we identify that this distillation objective exhibits recursive substructure—the optimal step schedule for $N$ steps inherently contain optimal subschedules for fewer steps. Exploiting this insight, we develop a dynamic programming algorithm that systematically derives theoretically optimal step sequences, ensuring maximal proximity to the teacher model's output under any pre-defined student sampling steps. It guarantees that the distilled schedule achieves the closest approximation to the teacher's continuous trajectory under computational constraints.

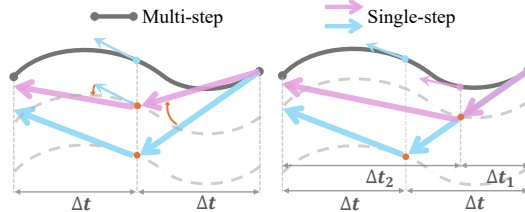

Figure 2: Two key factors in diffusion sampling: direction strategy(left) and stepsize strategy(right).

Importantly, our experimental analysis demonstrates the universal applicability and robustness of the proposed stepsize distillation framework. It exhibits consistent performance across various model architectures (e.g., U-Net Karras et al. (2022) vs. Transformer-based Peebles & Xie (2023)), ODE solver orders (1st to 3th-order Lu et al. (2022b)), noise schedules (linear Song et al. (2020a)/Flow Matching Liu et al. (2022b); Lipman et al. (2022)/EDM Karras et al. (2022)), and diverse denoising direction strategies Zhao et al. (2024); Lu et al. (2022a;b). Furthermore, when applying the optimal stepsize to text2image and text2video diffusion pipelines, our method achieves **$10\times$** speedups while preserving **99.4%** of the teacher model's performance metrics on GenEval Ghosh et al. (2023) benchmark. Such universal acceleration capability suggests broad practical potential for deploying latency-efficient diffusion models without sacrificing output quality.

Above all, our key contributions are as follows:

1. Theoretically Optimal Stepsize Distillation: We formulate stepsize optimization as a dynamic programming problem, proving that our solution can achieve global error minimization.

2. Architecture-Agnostic Robustness: We demonstrate that the distilled schedules generalize across datasets, noise schedules, and ODE solvers, outperforming heuristic baselines in diverse settings.

3. Efficient Adaptation: Our method enables lightweight schedule calibration across tasks, achieving 5-10$\times$ speedup with negligible performance degradation.

| Criterion | AYS Sabour et al. (2024) | GITS Chen et al. (2024) | DM Xue et al. (2024) | LD3 Tong et al. (2024) | OSS |
|---|---|---|---|---|---|
| Backprop-free | ✓ | ✓ | ✓ | ✗ | ✓ |
| Stability | ✗ | ✓ | ✓ | ✓ | ✓ |
| Solver-aware | ✓ | ✗ | ✗ | ✓ | ✓ |
| Global error | ✗ | ✗ | ✗ | ✓ | ✓ |

Table 1: Related work comparison. Backprop-free denotes whether eliminating backpropagation during optimization. Stability denotes whether requires specialized stabilization techniques. Solver-aware means the algorithm integrates solver dynamics and network states. Global error means optimizing the final calibration error.

## 2 RELATED WORK

### 2.1 DIFFUSION SAMPLING DIRECTION STRATEGY

Recent advances focused on correcting high-order direction for diffusion sampling acceleration. The earliest work Song et al. (2020a) adopted first-order Euler for discretization diffusion ODEs. EDM Karras et al. (2022) pioneers adopting Heun'sAscher & Petzold (1998) method for second-order gradient estimation between steps, reducing the Euler approximation error. PNDM Liu et al. (2022a) figured out the way of solving the denoising differential equation on manifold. AMED Zhou et al. (2024) utilized an additional network to predict the middle point of each step in the second order Heun solver. DPM-Solver Lu et al. (2022a) leveraged the semi-linear structure of diffusion ODEs, decomposing backward processes into linear/nonlinear components through Taylor expan-

sion. Building on this, DPM-Solver++ Lu et al. (2022b) introduced a multistep gradient reuse and shifts schedule, further saving the computational efforts. UniPC Zhao et al. (2024) generalized these via a predictor-corrector framework using Lagrange polynomial interpolation, enabling arbitrary-order accuracy with minimal computational overhead. These innovations collectively established theoretical foundations for balancing discretization precision and computational overhead.

## 2.2 DIFFUSION SAMPLING STEPSIZE STRATEGY

Traditional diffusion models typically employ predefined fixed-step strategies. For instance, DiT Peebles & Xie (2023) and SDXL Podell et al. (2023) adopt uniform timestep allocations, while EDM Karras et al. (2022) and CM Song et al. (2023) rely on manually crafted schedules. These strategies often yield suboptimal results in few-step sampling scenarios, prompting recent studies to explore stepsize optimization.

Existing methods fall into two categories: training-free optimization and learning-based strategies. The former approaches, including DM Xue et al. (2024), which minimizes inference error bounds via constrained trust-region methods, have a poor generalization ability since it shares the stepsize strategy across different diffusion networks and different solvers. Similarly, AYS Sabour et al. (2024) optimizes stepsize through KL divergence upper bounds (KLUB) but remains confined to SDE frameworks. Jolicoeur-Martineau et al. (2021) gets a better stepsize according to the similarity with the results of the high-order solver, which mixes the direction and stepsize of denoising. Watson et al. (2021b) selects the sampling steps based on dynamic programming and minimizes the sum of ELBO but incurs significant performance degradation. GITS Chen et al. (2024) also employs dynamic programming to optimize stepsize but exhibits deviations from minimizing global error due to fixed cost matrices. The latter category, exemplified by LD3 Tong et al. (2024), leverages a monotonic network for stepsize prediction, which is supervised by global errors. It based on the timestep prediction network, which is different from our DP based method. GGDM Watson et al. (2021a) adds trainable parameters to the schedule and achieves high-quality samples through well-designed training. We summarize some recent methods in Table 1 from different perspectives.

## 3 METHOD

### 3.1 DIFFUSION SAMPLING PRELIMINARIES

The continuous diffusion model progressively adds Gaussian noise to a data point $x_0$ in the forward process:

$$q(x_t|x_0) = \mathcal{N}(x_t|\alpha_t x_0, \sigma_t^2 I) \tag{1}$$

where $\lambda_t = \frac{\alpha_t^2}{\sigma_t^2}$ denotes the Signal-to-Noise-ratio (SNR). The denoising process generates images from either Stochastic Differential Equation (SDE) solvers Ho et al. (2020); Nichol & Dhariwal (2021) or Ordinary Differential Equation(ODE) solvers Liu et al. (2022b); Song et al. (2020a). The key difference between them is whether there is randomness in the generation process. Since ODE eliminates randomness, it often achieves significantly better results with fewer sampling steps. Therefore, this paper mainly focuses on the ODE solvers that follows:

$$dx_t = [f_t x_t - \frac{1}{2}g_t^2 \nabla_x log p_t(x_t)]dt \tag{2}$$

where $f_t = \frac{dlog\alpha_t}{dt}$ and $g_t^2 = \frac{d\sigma_t^2}{dt} - 2\frac{dlog\alpha_t}{dt}\sigma_t^2$. The generation process essentially involves solving the ODE and integrating the following expression from timestep $T$ to 0:

$$x_0 = x_T + \int_T^0 f_t x_t - \frac{1}{2}g_t^2 \nabla_x log p_t(x_t)dt \tag{3}$$

In practical applications, we usually discretize it into a finite number of $N$ steps to speed up the inference process:

$$x_0 = x_T + \sum_{i=N}^{1} v_{\theta,i}(t_{i-1} - t_i) \tag{4}$$

where $\boldsymbol{x_T} \sim N(0,1)$ is the initial noise, $\boldsymbol{\theta}$ denotes the parameter of denoising model, $\boldsymbol{v_{\theta,i}}$ denotes the optimization direction on step $i$. For simplicity, we use $\boldsymbol{v_\theta}$ to represent the direction strategy. $t_{i-1} - t_i$ denotes the stepsize on each optimization step. Many previous works have explored the selection of optimization direction $\boldsymbol{v_i}$. For example, in Heun Ascher & Petzold (1998); Karras et al. (2022), $\boldsymbol{v_i} = \frac{1}{2}(\boldsymbol{v}_{t_i} + \boldsymbol{v}_{t_{i+1}})$, in DPM-Solver Lu et al. (2022a), $\boldsymbol{v_i} = \boldsymbol{v}_{t_\lambda(\frac{\lambda_{t_i} + \lambda_{t_{i+1}}}{2})}$, $t_\lambda$ is the inverse function mapping from $\lambda$ to $t$.

However, few works have discussed the selection of stepsize $t_i$. Most of the previous works can be divided into two categories. The first class selects a uniform $t_i$:

$$t_i = \frac{i}{N} \quad where\ i = 0 \ldots N \tag{5}$$

such practices include LDM Rombach et al. (2022), DiT Peebles & Xie (2023), FLUX Labs (2023), SDXL Podell et al. (2023), etc. The second class includes EDM Karras et al. (2022), CM Song et al. (2023), which adopts a handcraft stepsize schedule:

$$t_i = (t_1^{1/\rho} + \frac{i-1}{N-1}(t_N^{1/\rho} - t_1^{1/\rho}))^\rho \tag{6}$$

However, these ad-hoc schedules always lead to a sub-optimal result. In this paper, we emphasize that the choice of stepsize is particularly crucial, especially in scenarios with limited steps. Besides, the stepsize operates orthogonally to the directional strategy.

### 3.2 DEFINITION OF OPTIMAL DENOISING STEPSIZE

Although discretizing the ODE trajectory can significantly increase the speed, it will deviate from the original ODE and cause the results to deteriorate to a certain extent. Therefore, the stepsize discretization aims to generate results that are as consistent as possible with the non-discretization. This is similar to traditional distillation, so we formulate our task as stepsize distillation with the difference being that no parameter training is required.

Suppose that we have a teacher inference schedule, including both the direction schedule ($\boldsymbol{v_\theta}$ in Equation 4) and the stepsize schedule ($\{t^N\} = \{t_N, t_{N-1}, ..., t_1, t_0\}$), where the number of steps $N$ is large enough to approximate infinite steps sampling results. We abbreviate the result generated from the initial point $\boldsymbol{x_T}$ through the stepsize strategy $\{t^N\}$ and direction strategy $\boldsymbol{v_\theta}$ according to Equation 4 as $\mathcal{F}(\boldsymbol{x_T}, \boldsymbol{v_\theta}, \{t^N\})$. For one step denoising of state $\boldsymbol{x_i}(t_j < t_i)$ starts from $t_i$ and ends at $t_j$, the results can be format as follows:

$$\mathcal{F}(\boldsymbol{x_i}, \boldsymbol{v_\theta}, \{t_i, t_j\}) = \boldsymbol{x_i} + \boldsymbol{v_\theta}(t_j - t_i) \tag{7}$$

We aim to distill a student stepsize schedule, ($\{s^M\} = \{s_M, s_{M-1}, ..., s_1, s_0\}$), where $\{s^M\}$ is a subset of $\{t^N\}$ and $M$ is smaller than $N$. For a fixed initial noise $\boldsymbol{x_T}$, the optimal student stepsize schedule $\{s^M\}^o$ aims to minimize the global discretization error with the teacher generation result:

$$\{s^M\}^o = \underset{\{s^M\} \subset \{t^N\}}{\arg\min} \left\| \mathcal{F}(\boldsymbol{x_T}, \boldsymbol{v_\theta}, \{t^N\}) - \mathcal{F}(\boldsymbol{x_T}, \boldsymbol{v_\theta}, \{s^M\}) \right\|_2^2 \tag{8}$$

### 3.3 DYNAMIC PROGRAMMING BASED CALIBRATION

#### 3.3.1 RECURSIVE SUBTASKS

Our goal is to use $M$ student steps to approximate $N$ teacher steps sampling results as closely as possible. We separate it into subtasks: *using $i$ student steps to approximate $j$ teacher steps sampling results*.

Therefore, we make such notations: the sampling trajectory of the teacher model is $\{\boldsymbol{X^t}\} = \{\boldsymbol{x}[t_N], \boldsymbol{x}[t_{N-1}], ..., \boldsymbol{x}[t_1], \boldsymbol{x}[t_0]\}$ based on the stepsize schedule $\{t^N\}$, where $\boldsymbol{x}[t_j] = \mathcal{F}(\boldsymbol{x_T}, \boldsymbol{v_\theta}, \{t_N, ..., t_j\})$ denotes the sampling result at timestep $t_j$. For simplicity, we abbreviate $\boldsymbol{x}[t_j]$ as $\boldsymbol{x}[j]$. For the student model, we denote the optimal denoised result using $i$ step to

approximate $\boldsymbol{x}[j]$ as $\boldsymbol{z}[i][j]$. Moreover, We additionally record that $\boldsymbol{z}[i][j]$ comes from timestep $r[j](r[j] > j)$, which means:

$$r[j] = \underset{j+1,...,N}{\arg\min} \|\boldsymbol{x}[j] - \mathcal{F}(\boldsymbol{z}[i-1][r[j]], \boldsymbol{v_\theta}, \{r[j], j\})\|_2^2 \tag{9}$$

We can find that these subtasks demonstrate a Markovian recursive property contingent upon the number of student steps utilized: The student model's $i$-th step approximation for the teacher's $j$-th step is determined by its prior result, where the approximation at timestep $r[j]$ utilized $i-1$ denosing steps. It can be formally expressed as:

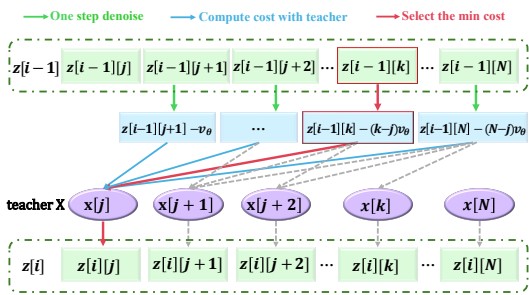

$$\boldsymbol{z}[i][j] = \mathcal{F}(\boldsymbol{z}[i-1][r[j]], \boldsymbol{v_\theta}, \{r[j], j\}) \tag{10}$$

By combining Equation 10 and Equation 7, the recursion relation can be written as:

Figure 3: Subtask illustration of the recursive subtasks. The optimal results at timestep $j$ using $i$ step denosing ($z[i][j]$) derives from the $i-1$ step optimal denosing results($z[i-1]$).

$$\boldsymbol{z}[i][j] = \boldsymbol{z}[i-1][r[j]] + \boldsymbol{v_\theta}(j - r[j]) \tag{11}$$

The illustration of solving the recursive subtasks is shown in Figure 3.

### 3.3.2 DYNAMIC PROGRAMMING

According to the recursion formula, we use dynamic programming to solve the stepsize selection problem for the student model. This process can be regarded as searching $\boldsymbol{z}[i][j]$ and recording each $r[j]$ as $i$ increases. Then we can obtain the final result of using $M$ student steps approximating $N$ teacher steps $\boldsymbol{z}[M][N]$. Finally, we backtrack the timestep $r[j]$ as the optimal trajectory. In this way, we distilled the optimal stepsize of the student model from the teacher steps. The pseudo code is provided in Algorithm 1.

### 3.3.3 THEORETICALLY OPTIMAL STEPSIZE

To rigorously establish the theoretical optimality of the dynamic programming algorithm in attaining the optimal stepsize schedule, we demonstrate that this subtask decomposition simultaneously satisfies both *ovealapping subproblems* and *optimal substructure*.

**Overlapping Subproblems**. This can be easily found since each denoised step is only dependent on the previous one step result, which is independent of the denoised history before. Take the one order solver as an example. Suppose we want to denoise from timestep $s$ to timestep $t$, we have:

$$\boldsymbol{x_t} = \boldsymbol{x_s} + \boldsymbol{v_\theta}(t - s) \tag{12}$$

One special case is the multistep high order solvers such as DPM-Solver Lu et al. (2022b) and UniPC Zhao et al. (2024). Compared with one order solver, the only difference is that the denoising process needs to rely on multi-step results. In this case, the overlapping subproblem changes into denoising from multiple states with different noise intensities. We leave the details about searching with high order solvers in the Appendix.

**Optimal Substructure**. The optimal substructure enables pruning suboptimal denoising trajectories in advance by dynamic programming. Here, we illustrate this substructure in Lemma 3.1 and leave the proof in the Appendix.

**Lemma 3.1.** *The optimal m step denosing results of student solver $z[m]$ always derives from the optimal m-1 step results $z[m-1]$ with additional one step denosing.*

### 3.4 AMPLITUDE CALIBRATION

While our step optimization enables efficient trajectory approximation, we identify a critical challenge when applying few-step inference: systematic amplitude deviation becomes pronounced in low-step regimes. In Figure 4, we plot the the quantile of 5% and 95% percentiles of the input tensor to denoising models across varying noise intensities. We find that the distribution changes significantly between the few-step (5) and multiple-step (200) trajectories, especially in the later stages of the denoising process.

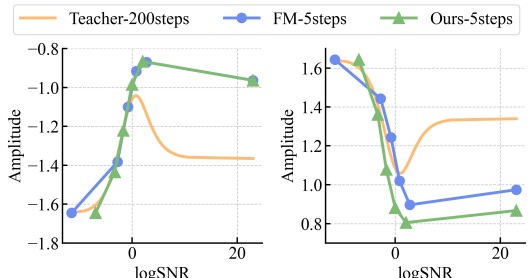

Figure 4: Amplitude of input tensor throughout denoising steps. The left and right plot the quantile of 5%, and 95% respectively.

To address this, we propose a per-step affine transformation that aligns student outputs with teacher amplitude characteristics. For each noise level $\lambda_t$, we calibrate the transformation using a linear function:

$$\hat{\boldsymbol{x}}_t = (S_{95} - S_5)/(x_{t,95} - x_{t,5})\boldsymbol{x}_t \tag{13}$$

where $S_{95}$ and $S_5$ denote the quantile ranges of 95%-5% of the teacher on 512 samples, which are calculated in advance. $x_{t,95}$ and $x_{t,5}$ are the quantile ranges of 95% - 5% of the student at step $t$ during denosing. In Section 4.2.1, we show that applying this calibration at the test time can significantly improve details such as color and texture.

## 4 EXPERIMENTS

**Implementation details.** Our algorithm features plug-and-play compatibility, allowing for seamless integration into various diffusion frameworks with minimal adaptation effort. To ensure consistent parameterization across different implementations, we adopt the velocity prediction paradigm from Flow Matching(FM)Liu et al. (2022b); Lipman et al. (2022) as our unified inference target. This requires wrapping the existing models with two standardization operations: (1) reparameterize the prediction target to velocity space, and (2) align the sampling trajectory based on the Signal-to-Noise ratio (SNR) matching. As theoretically guaranteed by Lee et al. (2024), this standardization does not affect the sampling result while enabling unified stepsize optimization across architectures. In the Appendix, we will explain this unification in detail.

**Evaluation Protocol** We prioritize minimizing the divergence of the trajectory from the teacher models as our main optimization objective. For quantitative evaluation, we employ PSNR to measure pixel-level fidelity between student and teacher output. Unless otherwise specified, all teacher models utilize 200 sampling steps to empirically balance computational efficiency with output quality across most test domains.

### 4.1 ROBUSTNESS ANALYSIS

We conduct comprehensive ablation studies to validate our method's robustness across four critical dimensions: noise schedules (Section 4.1.1), number of teacher steps (Section 4.1.2), ODE solver orders (Section 4.1.3), and diverse modeling frameworks (Section 4.1.4).

#### 4.1.1 ROBUSTNESS TO NOISE SCHEDULES

To evaluate schedule invariance, we performed class-conditional image generation experiments on ImageNet-64Russakovsky et al. (2015) using an EDM-based Karras et al. (2022) diffusion model. We test three established noise schedules as teacher configurations:

- EDM's exponential noise schedule Karras et al. (2022).

- DDIM's linear noise schedule Song et al. (2020a).

- Flow Matching's schedule Lipman et al. (2022)

Each teacher generates reference samples using 200 steps. We compare our optimized steps against naive uniform step reduction, measuring similarity through PSNR between student and teacher outputs. As shown in Section 2, our method achieves 2.65 PSNR improvements on average over baseline step reduction across all schedules, demonstrating consistent performance regardless of the teacher's schedule.

|  | steps | EDM | DDIM | FM |
|---|---|---|---|---|
| Naive step reduction | 50 | 33.71 | 34.31 | 31.78 |
|  | 20 | 25.76 | 26.12 | 23.97 |
|  | 10 | 20.83 | 20.89 | 19.00 |
|  | 5 | 16.38 | 15.96 | 15.69 |
| OSS | 20 | 28.21(**+2.45**) | 28.07(**+1.95**) | 28.51(**+4.54**) |
|  | 10 | 22.88(**+2.05**) | 22.79(**+1.9**) | 23.02(**+4.02**) |
|  | 5 | 18.41(**+2.03**) | 18.05(**+2.09**) | 18.53(**+2.84**) |

Table 2: ImageNet-64 pixel space conditional image generation with different teacher schedules. All the student solvers are learned from the corresponding 200-step teacher schedule. Results are shown in PSNR calculated with 200-step corresponding teacher results.

### 4.1.2 ROBOSTNESS TO NUMBER OF STEPS

We analyze the impact of the number of teacher steps based on ImageNet experiments. The teacher solver adopts the DDIM schedule and generates images using various numbers of steps from 100 to 1000 (student fixed at 20 steps). We measure the PSNR against 1000-step teacher results. Table 3 reveals that the performance basically converged in 200 steps. This indicates that the 200-step teacher model already provides an adequate search space for the dynamic programming algorithm to identify the optimal 20-step policy. Moreover, our method also maintains fidelity well, even when the teacher model has only 100 steps.

We further analyze the variance of step distribution across samples in Table 5. For teacher solvers with different steps, the optimal step sequences almost overlap. Besides, different samples demonstrate minor differences (light purple area in the figure). This allows us to generalize to unsearched samples by averaging over a selected subset of instances, thereby reducing the inference overhead in a zero-shot manner. We named this setting as *OSS-ave*. Specifically, by employing 512 samples to compute the mean of the optimal stepsize schedule, we achieve results that closely approximate those obtained through instance-specific optimization, as shown in Table 3.

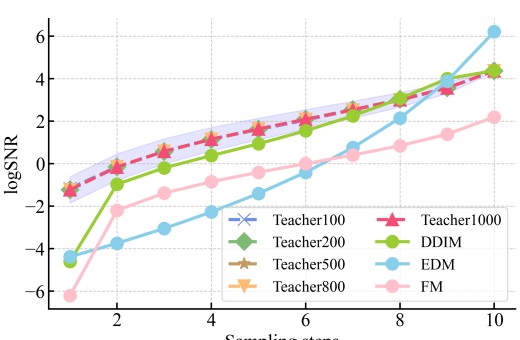

Figure 5: Step schedule for different solvers. Our method achieves nearly identical results from different teacher model steps.

| Teacher steps | 100 | 200 | 500 | 800 | 1000 |
|---|---|---|---|---|---|
| DDIM teacher | 37.97 | 44.24 | 52.93 | 57.30 | - |
| OSS | 27.04 | 28.07 | 27.11 | 27.12 | 27.13 |
| OSS-ave | 26.41 | 26.47 | 26.47 | 26.48 | 26.48 |

Table 3: ImageNet-64 pixel space conditional image generation results with different teacher steps. All the teacher solver leverages DDIM schedules. Students adopt 20-steps for inference. Results are shown in PSNR calculated with the 1000-step teacher.

### 4.1.3 ROBOSTNESS TO HIGHER ORDER SOLVER

Our optimal stepsize searching algorithm can be applied to higher-order solvers. We validate it using DiT-XL/2 modelPeebles & Xie (2023) on ImageNet 256×256 using classifier-free guidance

as 1.5. Figure 9a demonstrates that our step sequences combined with third-order solvers achieves significant improvements over the baseline DDIM schedule.

This verifies that optimal step scheduling and the solver order constitute complementary axes for few-step sampling improvement, enables practical acceleration without quality degradation.

### 4.1.4 GENERALIZATION ACROSS FRAMEWORKS

We validate framework-agnostic applicability through two different extensions: **Masked Autoregressive (MAR) Generation:** By applying the optimal stepsize schedule to the MAR Li et al. (2025) ImageNet-256 model, we reduce the sampling steps from 500 to 5, achieving $100\times$ acceleration while maintaining competitive performance (FID=4.67). When decreasing steps to 50, the FID remained nearly unchanged (2.66→2.59), demonstrating robust stability. Note that the model MAR-Base contains 32 tokens, but we only search for the first token and apply the result steps to the other 31 tokens, which further demonstrates the robustness. The result are

**Video Diffusion:** In Open-Sora Zheng et al. (2024) frameworks, our optimized schedules enable $10\times$ speedups while preserving visual fidelity. Visualization results are shown in the Appendix. We find that simply reducing the sampling step to 20 steps completely changes the generated video. However, by adjusting the sampling steps, the generated result is almost the same as the teacher with $10\times$ speedup.

### 4.2 EXTENSIONS OF OPTIMAL STEPSIZE

### 4.2.1 AMPLITUDE CALIBRATION

As demonstrated in Section 3.4, insufficient sampling steps may induce significant amplitude discrepancies between generated intermediate states and the teacher model's trajectories. To investigate it, we conducted qualitative analysis on the ImageNet dataset. Although amplitude adjustment may slightly decrease the PSNR compared to the teacher model's outputs (18.53 → 16.92), the adjusted samples exhibit enhanced structural detail and demonstrate improved realism as evidenced in Figure 6.

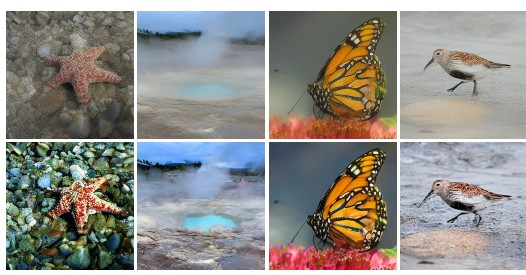

Figure 6: Through amplitude adjustment, the synthesized outputs (bottom row) exhibit enhanced details.

| Method | steps | Overall | Single object | Two object | Counting | Colors | Position | Color attribution |
|---|---|---|---|---|---|---|---|---|
| Flow matching | 100 | 0.649 | 0.989 | 0.803 | 0.722 | 0.774 | 0.220 | 0.390 |
| Flow matching | 10 | 0.590 | 0.963 | 0.727 | 0.663 | 0.668 | 0.213 | 0.310 |
| DDIM | 10 | 0.623 | 0.971 | 0.742 | 0.694 | 0.758 | **0.22** | 0.353 |
| GITS | 10 | 0.604 | **0.981** | 0.732 | 0.678 | 0.697 | 0.210 | 0.325 |
| DM | 10 | 0.643 | 0.971 | **0.788** | 0.719 | **0.777** | 0.188 | **0.415** |
| OSS-ave | 10 | **0.645** | **0.981** | 0.775 | **0.728** | **0.777** | 0.195 | **0.415** |

Table 4: Results of Geneval based on the Flux.1-dev model with different sampling schedules.

### 4.2.2 OPTIMAL STEPSIZE AS BALANCED TASK PARTITION

Our optimal stepsize schedule not only enables directly accelerating diffusion models sampling without finetuning network, but also inherently reflects the varying difficulty levels of denoising tasks across different noise intensities. This observation suggests that the optimal stepsize schedule can be interpreted as a principled multitask partitioning strategy, which may better balance the subtask relationships compared to conventional uniform timestep division, thereby mitigating distortion on final outputs. To validate this, we conduct experiments on ImageNet generation with the DiT-XL/2 Peebles & Xie (2023) model by splitting the entire denoising process into five consecutive stages, with each stage modeled by independent parameters. Inspired by Salimans & Ho (2022);

A photo of a yellow dining table and a pink dog

A photo of a cow left of a stop sign

A photo of a toaster below a traffic light

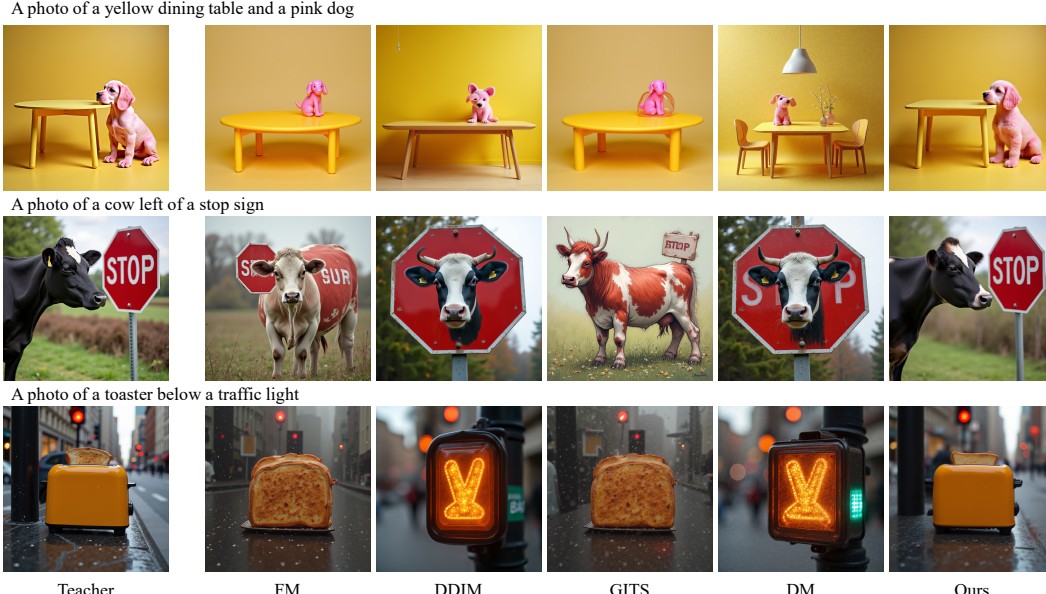

| Teacher | FM | DDIM | GITS | DM | Ours |

Figure 7: Visualization results on Geneval benchmark. Our optimal sampling schedule can produce results that are more similar to multi-step teachers, inherited strong instruction following ability.

Balaji et al. (2022), all sub-models are initialized with the pre-trained teacher model to accelerate convergence. The results shown in Table 7 in the appendix demonstrate that our optimal stepsize can be regarded as a good task division, better allocating the model capacity. It can be further improved when combined with other techniques, such as progressive distillation Salimans & Ho (2022).

### 4.3 COMPARE WITH OTHER METHODS

We compare our method with other stepsize strategies including ad-hoc schedules like DDIM, EDM, and flow matching, as well as dynamic adjustment methods such as DM Xue et al. (2024) and GITS Chen et al. (2024). Table 5 demonstrates our experimental results across four distinct datasets. Our method achieves results closest to those of the teacher model under 5 and 10 step generation setting. Moreover, we also validate our results on the GenEval Ghosh et al. (2023) benchmark using Flux.1-dev Labs (2023) as the foundation model. We list the quantitative results in Table 4, where the OSS-ave setting adopts 32 images for each category. We find that the optimal stepsize schedule achieves $10\times$ acceleration compared to the 100-step teacher model with almost no performance degradation on the GenEval benchmark. As shown in Figure 7, our method demonstrates the closest proximity to the teacher model comparing with other approaches.

| Method | 5steps | 10steps | 5steps | 10steps |
|---|---|---|---|---|
| | ImageNet-64 | | LSUN-Bed-256 | |
| EDM | 17.72 | 22.33 | 15.60 | 19.61 |
| DDIM | 17.69 | 22.22 | 15.39 | 20.18 |
| FM | 17.80 | 22.47 | 14.43 | 18.23 |
| DM | 17.39 | 20.50 | **18.90** | 20.67 |
| GITS | 16.29 | 19.40 | 11.58 | 13.55 |
| OSS-ave | **17.94** | **22.47** | 16.51 | **21.25** |
| | FFHQ-64 | | MSCOCO-512 | |
| EDM | 18.68 | 23.24 | 13.43 | 15.45 |
| DDIM | 18.37 | 23.57 | 12.31 | 14.58 |
| FM | 17.72 | 21.31 | 11.90 | 13.51 |
| DM | 19.79 | 22.69 | 13.50 | 14.73 |
| GITS | 20.46 | 25.20 | 10.60 | 11.07 |
| OSS-ave | **20.56** | **25.21** | **14.10** | **16.45** |

Table 5: PSNR results of different stepsize schedules.

### 5 CONCLUSION

This paper proposes a dynamic programming framework for stepsize optimization in diffusion sampling. By reformulating stepsize scheduling as recursive approximation, our method derives theoretically optimal step sequences with low computational overhead. The experiments demonstrate universal robustness across architectures and solvers while maintaining output fidelity. Its seamless integration with existing direction optimization strategies enables practical deployment advantages. This work establishes an alternative pathway for efficient diffusion sampling.

ETHICS STATEMENT

Our work adheres to the ICLR Code of Ethics. The research relies on publicly available datasets, which were used in strict accordance with their licenses and terms of service. We acknowledge that our model, like other large-scale generative models, could potentially be misused to generate harmful, biased, or factually incorrect content. The primary goal of our research is to advance the understanding of unified MLLMs. We have not designed our model for any malicious purposes and advocate for the responsible development and deployment of this technology within the research community.

REPRODUCIBILITY STATEMENT

We are committed to the reproducibility of our research. All implementation details required to reproduce our results are provided within the paper and its supplementary materials. A detailed description of our methodology is presented in Section 3. The experimental setup, including key hyperparameters, training configurations, and evaluation protocols, is described in Section 4 and further elaborated in the Appendix. We have submitted our source code as supplementary material and will make it publicly available.

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

# APPENDIX

## A PROOF OF THE OPTIMAL SUBSTRUCTURE

Here, we aim to proof: *To solve the optimal approximation $\boldsymbol{z}[m][n]$, which indicate the $m$-step student outcome to the $n$-step teacher outcome, it always derives from $\boldsymbol{z}[m-1][k]$, where $k > n$.*

**Assumption 1.** *If $\boldsymbol{x}$ is closer to the teacher denosing result at $\boldsymbol{x}[i]$ than $\boldsymbol{x}'$, as the form $\|\boldsymbol{x} - \boldsymbol{x}[i]\|^2 \leq \|\boldsymbol{x}' - \boldsymbol{x}[i]\|^2$, we have:*

$$\|\boldsymbol{v}_{\boldsymbol{\theta},\boldsymbol{k}} - \boldsymbol{v}_{\boldsymbol{\theta}*}\|_2^2 \leq \|\boldsymbol{v}'_{\boldsymbol{\theta},\boldsymbol{k}} - \boldsymbol{v}_{\boldsymbol{\theta}*}\|_2^2 \tag{14}$$

We prove our theorem by the method of contradiction. Suppose there exist $\boldsymbol{x}'$ which is suboptimal than $\boldsymbol{z}[m-1][k]$ and satisfies:

$$\|\boldsymbol{x}' - \boldsymbol{x}[k]\|_2^2 \geq \|\boldsymbol{z}[m-1][k] - \boldsymbol{x}[k]\|_2^2 \tag{15}$$

But the optimal result at step $m$ is generated from the suboptimal result at timestep n, which satisfies:

$$\|\mathcal{F}(\boldsymbol{x}', \boldsymbol{v}_{\boldsymbol{\theta}}, \{k, n\}) - \boldsymbol{x}[n]\|_2^2 \tag{16}$$

$$\leq \|\mathcal{F}(\boldsymbol{z}[m-1][k], \boldsymbol{v}_{\boldsymbol{\theta}}, \{k, n\}) - \boldsymbol{x}[n]\|_2^2 \tag{17}$$

We use $\boldsymbol{v}'_{\boldsymbol{\theta}}$ represents the velocity of $\boldsymbol{x}'$ at timestep k, use $\boldsymbol{v}_{\boldsymbol{\theta}}$ represents the velocity of $\boldsymbol{z}[m-1][k]$ at timestep k:

$$\mathcal{F}(\boldsymbol{x}', \boldsymbol{v}_{\boldsymbol{\theta}}, \{k, n\}) = \boldsymbol{x}' + \boldsymbol{v}'_{\boldsymbol{\theta}}(n - k) \tag{18}$$

Also, we assume the teacher solver results are an excellent ground truth approximation. We use $\boldsymbol{v}_{\boldsymbol{\theta}*}$ represents the ground truth velocity, which follows:

$$\boldsymbol{x}[n] = \boldsymbol{x}[k] + \boldsymbol{v}_{\boldsymbol{\theta}*}(n - k) \tag{19}$$

*Proof.* Here, we prove this in two cases. When the $\boldsymbol{x}'$ and $\boldsymbol{x}$ are closed to the teacher $\boldsymbol{x}[k]$, we have:

$$\|\mathcal{F}(\boldsymbol{x}', \boldsymbol{v}_{\boldsymbol{\theta}}, \{k, n\}) - \boldsymbol{x}[n]\|_2^2 \tag{20}$$

$$= \|\boldsymbol{x}' - \boldsymbol{x}[k] + (n - k)(\boldsymbol{v}'_{\boldsymbol{\theta}} - \boldsymbol{v}_{\boldsymbol{\theta}*})\|_2^2 \tag{21}$$

$$\approx \|(n - k)(\boldsymbol{v}'_{\boldsymbol{\theta}} - \boldsymbol{v}_{\boldsymbol{\theta}*})\|_2^2 \tag{22}$$

$$\geq \|(n - k)(\boldsymbol{v}_{\boldsymbol{\theta}} - \boldsymbol{v}_{\boldsymbol{\theta}*})\|_2^2 \tag{23}$$

$$\approx \|\mathcal{F}(\boldsymbol{z}[m-1][k], \boldsymbol{v}_{\boldsymbol{\theta}}, \{k, n\}) - \boldsymbol{x}[n]\|_2^2 \tag{24}$$

$$\tag{25}$$

This is contradictory to the hypothesis Equation 16.

When the $\boldsymbol{x}'$ and $\boldsymbol{x}$ are far from the teacher $\boldsymbol{x}[k]$, the input of the denosing network is far away from the training input. Therefore, we consider vectors $\boldsymbol{x}' - \boldsymbol{x}[k]$ and $\boldsymbol{v}'_{\boldsymbol{\theta}} - \boldsymbol{v}_{\boldsymbol{\theta}*}$ to be statistically independent and treat them as random vectors in a high-dimensional space, thereby establishing their orthogonality. Then:

$$\|\mathcal{F}(\boldsymbol{x}', \boldsymbol{v}_{\boldsymbol{\theta}}, \{k, n\}) - \boldsymbol{x}[n]\|_2^2 \tag{26}$$

$$= \|\boldsymbol{x}' - \boldsymbol{x}[k] + (n - k)(\boldsymbol{v}'_{\boldsymbol{\theta}} - \boldsymbol{v}_{\boldsymbol{\theta}*})\|_2^2 \tag{27}$$

$$= \|\boldsymbol{x}' - \boldsymbol{x}[k]\|_2^2 + \|(n - k)(\boldsymbol{v}'_{\boldsymbol{\theta}} - \boldsymbol{v}_{\boldsymbol{\theta}*})\|_2^2 \tag{28}$$

$$+ 2 * \langle \boldsymbol{x}' - \boldsymbol{x}[k], (n - k)(\boldsymbol{v}'_{\boldsymbol{\theta}} - \boldsymbol{v}_{\boldsymbol{\theta}*}) \rangle \tag{29}$$

$$\approx \|\boldsymbol{x}' - \boldsymbol{x}[k]\|_2^2 + \|(n - k)(\boldsymbol{v}'_{\boldsymbol{\theta}} - \boldsymbol{v}_{\boldsymbol{\theta}*})\|_2^2 \tag{30}$$

$$\geq \|\boldsymbol{z}[m-1][k] - \boldsymbol{x}[k]\|_2^2 + \|(n - k)(\boldsymbol{v}_{\boldsymbol{\theta}} - \boldsymbol{v}_{\boldsymbol{\theta}*})\|_2^2 \tag{31}$$

$$\approx \|\mathcal{F}(\boldsymbol{z}[m-1][k], \boldsymbol{v}_{\boldsymbol{\theta}}, \{k, n\}) - \boldsymbol{x}[n]\|_2^2 \tag{32}$$

$$\tag{33}$$

---

**Algorithm 1** Optimal Stepsize Distillation.

$D_\theta$ : Denoising model which outputs the velocity.
$x[N]$, $z$ samples from $\mathcal{N}(0, 1)$
1: # get teacher trajectory.
2: **for** $i = N$ to 1 **do**
3:     $v = D_\theta(x[i], i)$
4:     $x[i - 1] = x[i]$ - $v$
5: # use the student i steps result to approach the teacher's j steps result.
6: **for** $i = 1$ to $M$ **do**
7:     $v = D_\theta(z[i - 1, :], range(N, 1))$
8:     $z_{tmp} = ones(N + 1, N) \times Inf$
9:     **for** $j$ in $N$ to 1 **do**
10:         $z_{tmp}[j, :] = z[i - 1, j] - v[j] \times range(0, j - 1)$
11:     **for** $j$ in $N - 1$ to 0 **do**
12:         $cost = MSE(z_{tmp}[:, j], x[j])$
13:         $r[i, j] = argmin(cost)$
14:         $z[i, j] = z_{tmp}[r[i, j], j]$
15: # retrieve the optimal trajectory.
16: $R = [0]$
17: **for** $i$ in $M$ to 1 **do**
18:     $j = R[-1]$
19:     $R$.append($r[i][j]$)
20: **return** $R[1 :]$

---

**Algorithm 2** Optimal stepsize for high order solver

$D_\theta$,$x[N]$,$z$ same as one order algorithm.
$od$: Inference Order.
$\mathcal{F}$ : Forward function, $\mathcal{F}(x_{cur}, t_{cur}, v, t_{next}, order)$
1: # get teacher trajectory $x[i]$
2: **for** $i = N$ to 1 **do**
3:     $v = D_\theta(x[i], i)$
4:     $x[i - 1] = \mathcal{F}(x[i], i, v, i - 1, od)$
5: # use the student i steps result to approach the teacher's j steps result.
6: **for** $i = 1$ to $M$ **do**
7:     $v = D_\theta(z[i - 1, :], range(N, 1))$
8:     $z_{tmp} = ones(od + 1, N + 1, N) \times Inf$
9:     **for** $o$ in $od$ to 1 **do**
10:         **for** $j$ in $N$ to 1 **do**
11:             $t_{next} = range(0, j - 1)$
12:             $z_{tmp}[o, j, :] = \mathcal{F}(z[i - 1, j], j, v[j], t_{next}, o)$
13:     **for** $j$ in $N - 1$ to 0 **do**
14:         $cost = MSE(z_{tmp}[:, :, j], x[j])$
15:         $r[i, j] = argmin(cost.flatten())$
16:         $z[k, i] = z_{tmp}[r[i.j]//(N + 1), r[i, j]\%(N + 1), j]$
17:         $r[i, j] = r[i, j]\%(N + 1)$
    # retrieve the optimal trajectory
18: $R = [0]$
19: **for** $i$ in $M$ to 1 **do**
20:     $j = R[-1]$
21:     $R$.append($r[i][j]$)
22: **return** $R[1 :]$

---

This is contradictory to the hypothesis Equation 16. Therefore, the original hypothesis is not valid, the optimal results always come from the previous optimal results, which is the optimal substructure of our searching problems. Algorithm details are shown in Algorithm 1     □

## B    SEARCHING STEPS WITH HIGH ORDER SOLVERS

Here, we introduce the algorithm of searching optimal steps for high order solvers. A key distinction lies in the solver's dynamic order selection capability during the search phase: at each optimization step, the student solver may execute operations corresponding to the specified denoising order or any lower-order approximations. The optimal configuration is selected through minimum-distance alignment with the teacher trajectory across all denoising orders. After searching, the student solver utilize the optimal steps with the order given before. The complete optimization procedure is formalized in Algorithm 2.

## C    UNIFICATION OF DIFFERENT SAMPLING TRAJECTORIES

This section elaborates on adapting various pre-trained models for sampling via flow matching. This process comprises three steps: transformation of network prediction targets, sampling trajectory alignment, and model input alignment. The network prediction targets, comprising $\epsilon$, $x_0$, $v$, exhibit mutual convertibility through reparameterization. Our framework unifies their conversion to velocity $v$ through systematic transformation.

Following the methodology in Lee et al. (2024), temporal alignment of sampling trajectories requires matching the signal-to-noise ratio (SNR) at each step. The flow matching diffusion process follows:

$$x_t = t\epsilon + (1 - t)x_0 \tag{34}$$

while we assume the conventional forward trajectory of the pre-trained model adheres to:

$$x_t = \alpha_t x_0 + \beta_t \epsilon \tag{35}$$

Temporal alignment between flow matching time $t$ and reference trajectory time $t'$ is established through the ratio constraint:

$$\frac{1 - t}{t} = \frac{\alpha_{t'}}{\beta_{t'}} \tag{36}$$

Furthermore, the network input alignment between the current trajectory position $x_t$ and the pre-trained model's expected input $x_{t'}$ is achieved via linear transformation:

$$x_{t'} = \frac{\alpha_{t'}}{1 - t} x_t \tag{37}$$

These aligned coordinates ($t'$ and $x_{t'}$) enable direct utilization of pre-trained diffusion models to estimate the current trajectory velocity at timestep $t$, thereby enables sampling through flow matching.

## D    VISUALIZATION OF OPTIMAL SAMPLING SCHEDULE

In this section, we present additional visualizations of sampling schedules. We maintain the original teacher schedule used in each open-source implementation: DiT Peebles & Xie (2023) employs DDIM Song et al. (2020a) schedule, while Open-Sora Zheng et al. (2024), Flux Labs (2023), and Wan WanTeam et al. (2025) adopt flow matching with timeshift Esser et al. as their default schedules. We compare different few-step sampling strategies and utilize the unified framework in Section C to align timesteps with the teacher model. We present the teacher model's sampling timesteps corresponding to each few-step sampling strategy in Figure 8.

## E    ABLATION OF METRIC FUNCTIONS

Conventional continuous diffusion alignment relies on MSE between teacher and student intermediate states $x_t$. However, MSE may induce over-smoothing in pixel-level generation tasks. We thus explore three metrics: (1) MSE on reparameterized clean data $x_0$, this can be regarded as SNR-weighted MSE on $x_t$ Hang et al. (2023); (2) Inception-V3 Szegedy et al. (2016) feature distance on reparameterized $x_0$; (3) perceptual distance utilizing the denoising network Peebles & Xie (2023) on intermediate feature $x_t$.

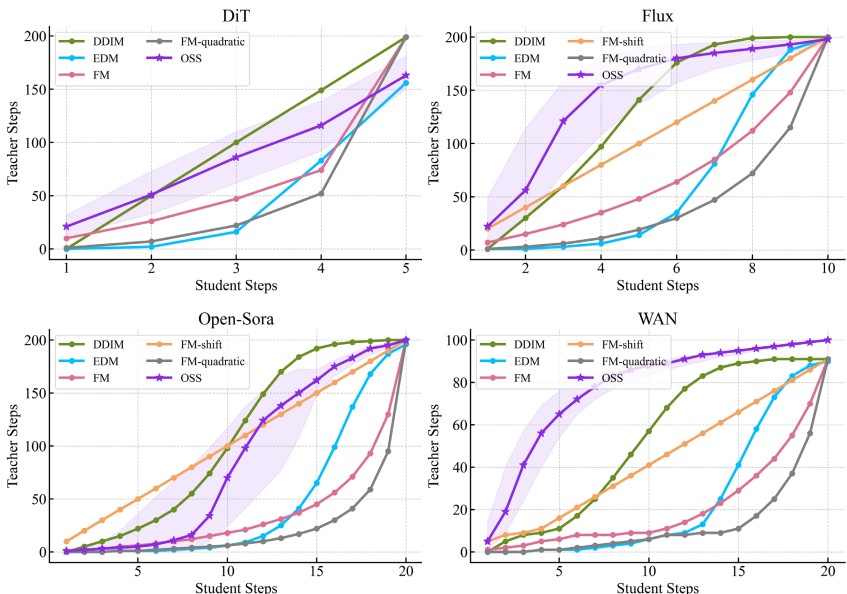

Figure 8: Different sampling schedules of DiT, Flux, Open-Sora, and WAN.

We evaluate the three aforementioned metrics on the ImageNet-64 generation with the pretrained EDM model and summarized the results in Table 6. The $x_0$-MSE setting outperforms the default distance metric calculated on $x_t$. However, the Inception-V3 feature metric performs worse, potentially attributed to lacking generalization ability on noisy data. Meanwhile, using the denoising model to calculate the perceptual distance does not demonstrate significant improvement.

| | $x_t$-MSE | $x_0$-MSE | $x_0$-IncepV3 | $x_t$-percep |
|---|---|---|---|---|
| PSNR ↑ | 18.54 | 22.45 | 9.91 | 18.27 |

Table 6: Ablation studies on different functions.

## F RESULTS OF DIFFERENT ORDER SOLVERS AND MAR-BASED

The results of different order solvers are shown in Figure 9a. The results of OSS under MAR-Base model are shown in Figure 9b.

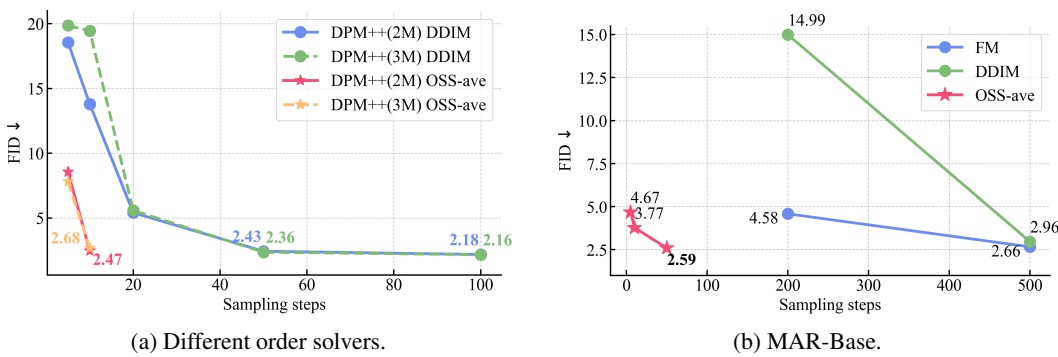

(a) Different order solvers.

(b) MAR-Base.

Figure 9: ImageNet-256 latent space conditional image generation results of different order solvers and MAR-Base.

|  | w/o training | w training |
|---|---|---|
| Uniform | 52.73 | 24.53 |
| OSS | 41.96 | **7.89** |

Table 7: Sampling results on ImageNet-256. Uniform is the schedule adopted in DDIM.

## G RESULTS OF TASK PARTITION BASED ON OSS.

## H MORE RESULTS

In this section, we provide more Flux.1-dev Labs (2023) generation results using 10 steps. Moreover, we also provide the video generation results in the github page.

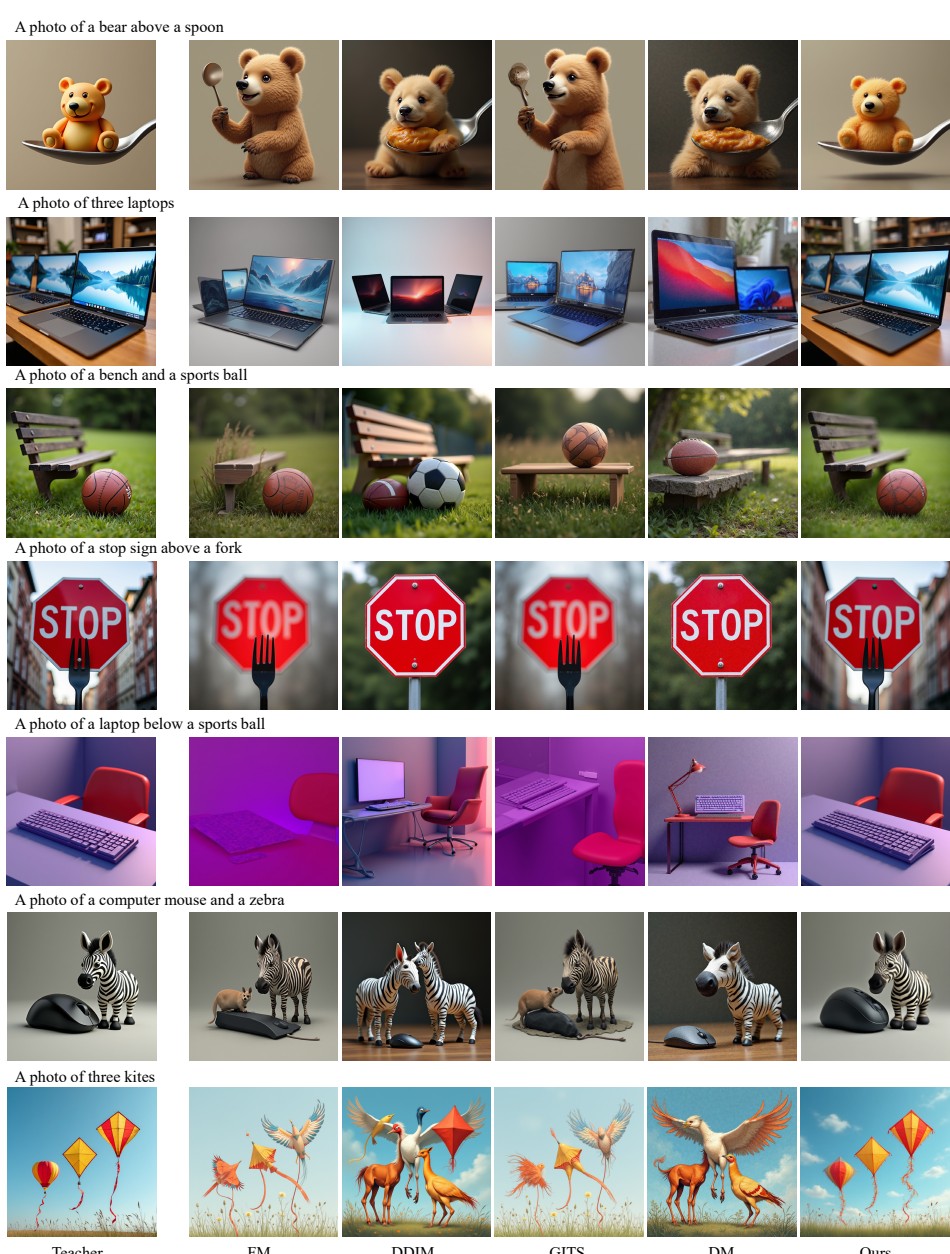

Figure 10: Flux generation results on Geneval.

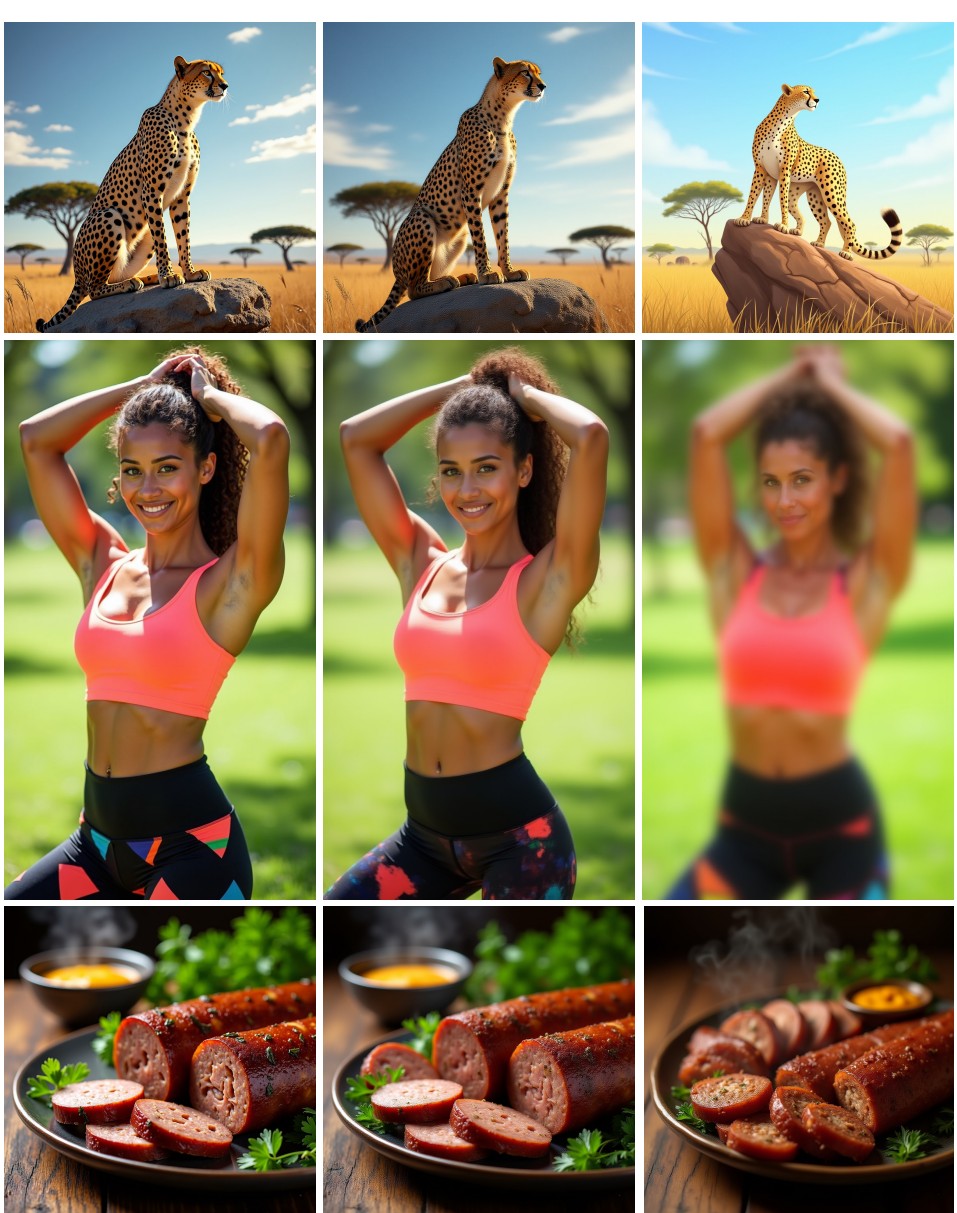

Figure 11: Flux generation results. Left: Original sampling result using 200 steps. Middle: Optimal stepsize sampling result within 10 steps. Right: Naively reducing sampling steps to 10.

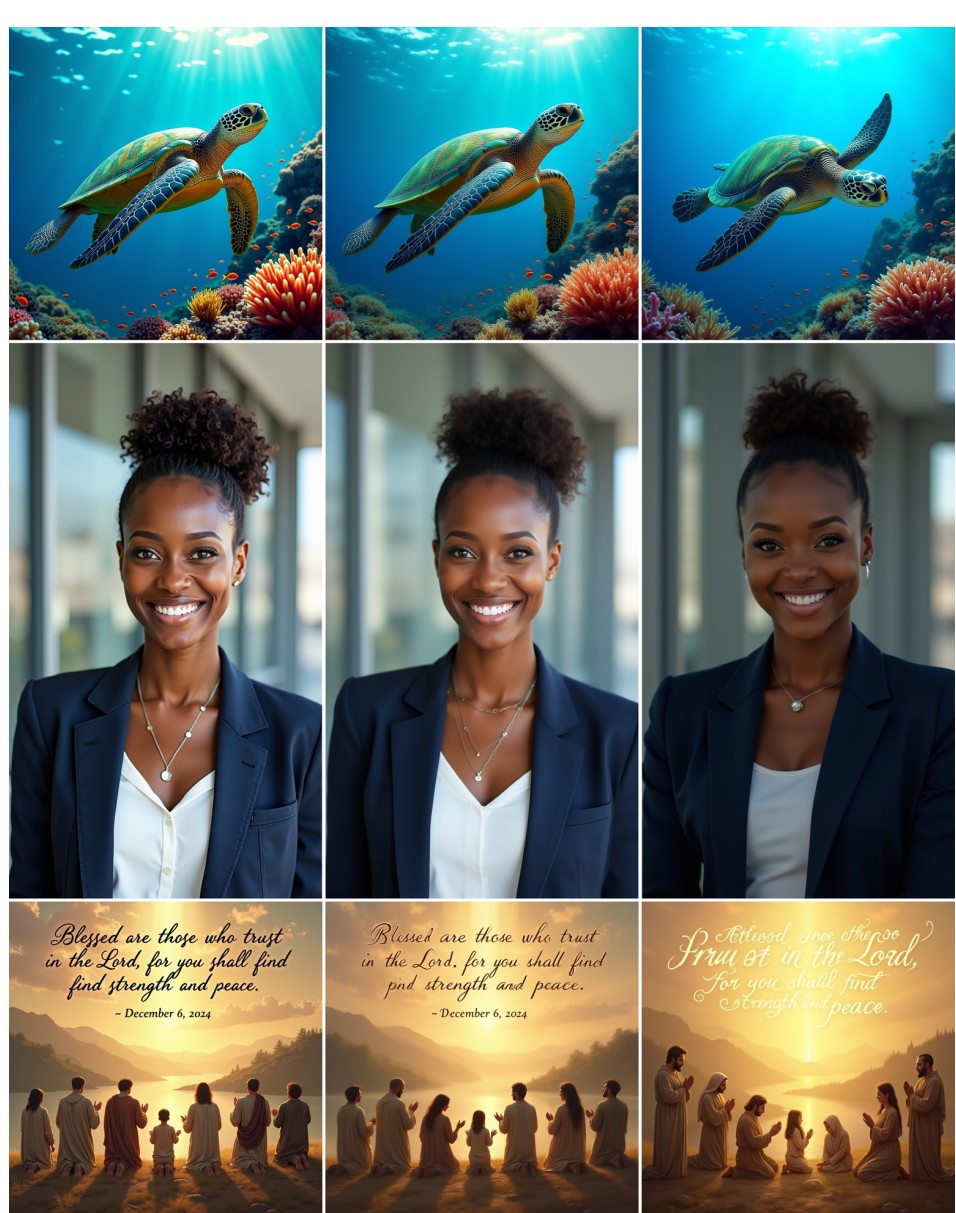

Figure 12: Flux generation results. Left: Original sampling result using 200 steps. Middle: Optimal stepsize sampling result within 10 steps. Right: Naively reducing sampling steps to 10.

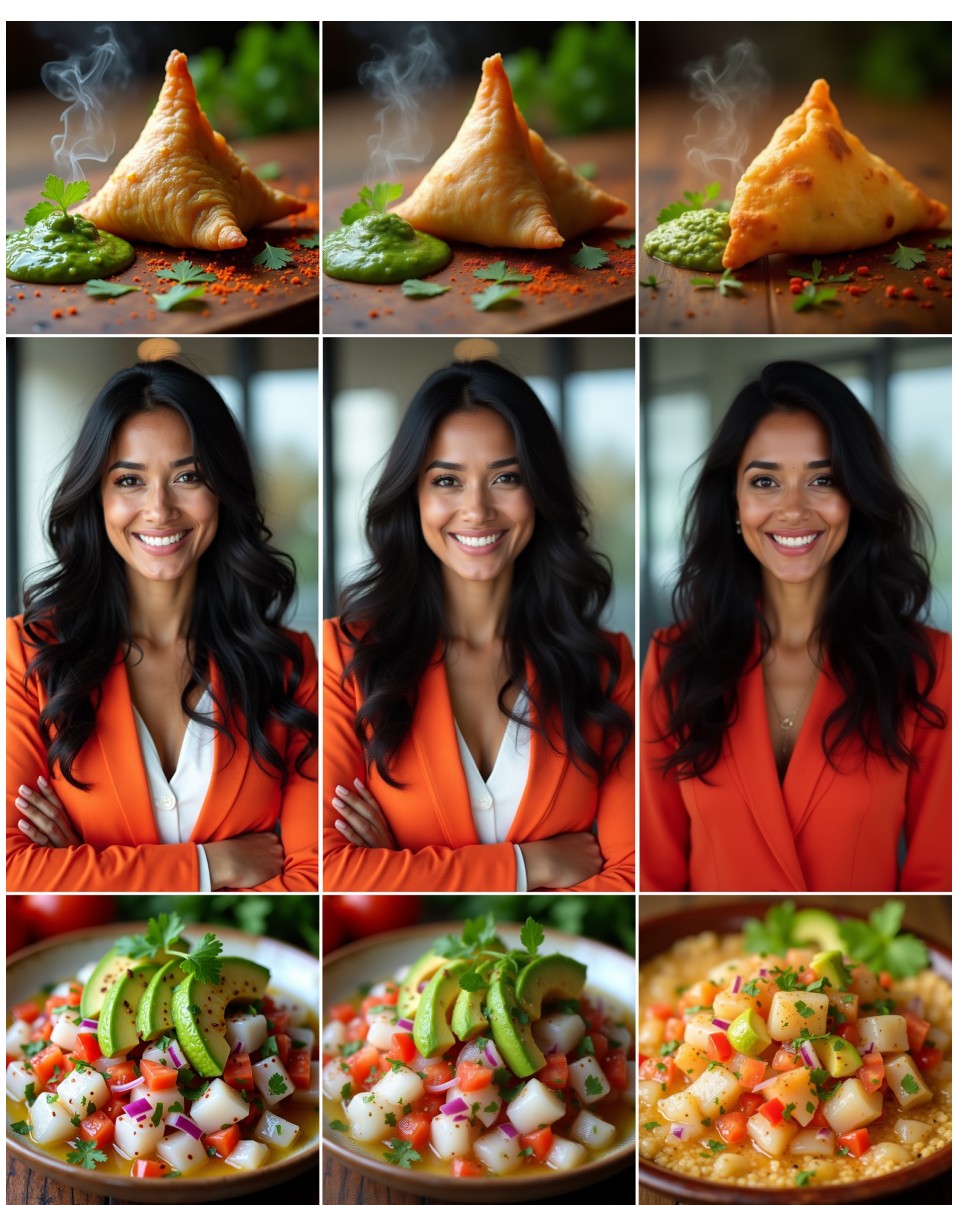

Figure 13: Flux generation results. Left: Original sampling result using 200 steps. Middle: Optimal stepsize sampling result within 10 steps. Right: Naively reducing sampling steps to 10.

