# OpenReview forum: "Optimal Stepsize for Diffusion Sampling"
_ICLR.cc/2026/Conference — Submitted to ICLR 2026_

### Official Review · Reviewer_YzEh · 2025-10-24

**Soundness:** 2
**Presentation:** 2
**Contribution:** 2
**Rating:** 4
**Confidence:** 4

**Summary:**

The paper proposes a dynamic programming-based framework, Optimal Stepsize Distillation (OSS), to optimize the step sizes (“stepsize schedule”) used in diffusion model sampling. Instead of relying on heuristic or hand-crafted stepsize choices, they recast the problem as a knowledge distillation task: a low-step (student) sampler aims to match the output of a high-step (teacher) sampler, minimizing global discretization error. The approach leverages the recursive structure of the sampling process and offers theoretically optimal solution via dynamic programming. OSS is shown to be architecture-agnostic and robust across different solvers, noise schedules, and tasks, enabling up to 10x acceleration in diffusion-based image and video generation with negligible loss in performance, as evidenced on the GenEval benchmark and other datasets. The paper provides extensive empirical results, detailed derivations, and justifies the approach with theoretical guarantees.

**Strengths:**

1. The recasting of stepsize optimization as a dynamic programming problem is well-motivated and rigorously constructed (see Section 3.3 and Lemma 3.1), with detailed equations and a clear algorithmic description (Algorithm 1 on pg. 14). The approach exploits the optimal substructure property, which is proven and illustrated.
2. The framework is general—applying to both first- and high-order ODE solvers (see Algorithm 2 in Appendix B, and main Section 3.3.3), and decouples stepsize schedule from direction choice.
3. Experiments are broad and robust, including ablation on noise schedules (Table 2, pg. 7), teacher stepsize granularity (Table 3, pg. 7), and evaluation across multiple datasets (Table 5, pg. 9). Figure 4 (pg. 6) illustrates how amplitude misalignment can be corrected by their calibration method—demonstrating engagement with practical sampling artifacts.

**Weaknesses:**

1. While the paper mentions several prior stepsize/schedule optimization works (e.g., GITS, DM, LD3), it does not clearly distinguish how OSS fundamentally surpasses or aligns with recent works such as DDSS [Watson et al., 2022] [1] and “Adaptive Time-Stepping Schedules for Diffusion Models” [Yuzhu et al., 2024] [2], both of which are missing from the reference and comparative experiments, and are strongly related in optimizing time steps or sampling schedules for improved efficiency. For instance, Section 2 does not articulate the improvement over gradient-based or convergence-bound approaches in these missing papers.
2. While the proof of optimal substructure (Appendix A) is mathematically sound under certain assumptions, the practical optimality is limited by (a) the granularity of teacher steps (as the student schedule is a subset of teacher steps), and (b) the potential mismatch between $L_2$ error and perceptual or application-specific metrics (see ablation Table 6, Appendix E). The claim of “global discretization bounds” should be qualified with these approximations and the fact that, in practice, optimality is with respect to the chosen metric (e.g., PSNR or MSE), not necessarily downstream sample quality.
3. The core algorithm minimizes $L_2$ distance between student and teacher trajectories (main Eq. 9, Alg. 1), but Table 6 (Appendix E) shows this choice can misalign with perceptual image metrics (e.g., Inception-V3 or feature distance). The methodology may lead to over-smoothing or suboptimal perceptual outcomes, as seen by the drop in PSNR after amplitude calibration (Section 4.2.1). This tradeoff between MSE/PSNR and perceptual detail is not thoroughly discussed or ablated.
4. While Figure 4 and Section 3.4/4.2.1 demonstrate the need for amplitude calibration, this fix is relatively ad hoc (a simple affine per-step transformation) and lacks a thorough theoretical or empirical analysis of its impact, especially on more challenging data distributions or with colored noise. There is no ablation on its robustness, generalizability, or risk of overfitting/failure modes.
5. While dynamic programming is, in theory, efficient, OSS appears to run a nontrivial search over a potentially large $N \times M$ table (Algorithm 1, Figure 3). There is no detailed discussion or timing comparison of search overhead versus gains in sampling time, nor a complexity analysis, especially for high-order solvers or for real-world large models.

[1] Watson D, Chan W, Ho J, et al. Learning fast samplers for diffusion models by differentiating through sample quality[C]//International Conference on Learning Representations. 2021.

[2] Chen Y, He F, Fu S, et al. Adaptive time-stepping schedules for diffusion models[C]//The 40th Conference on Uncertainty in Artificial Intelligence. 2024.

**Questions:**

1. How does OSS perform relative to gradient-based sampler optimization methods such as DDSS [Watson et al., 2022], both in terms of sampling quality (FID, IS) and search/compute overhead? Is there an empirical or conceptual advantage in practice?
2. Could the authors elaborate on the practical runtime/complexity tradeoff of the DP-based schedule search, especially for very deep teacher schedules or high-resolution images? Is the search cost amortized or negligible compared to model sampling cost?
3. Are there plans or methodology to extend OSS to adaptive (test-time) schedule computation, avoiding pre-computed mean sequence or per-sample search, to further increase practicality?

---

### Official Review · Reviewer_D3on · 2025-11-01

**Soundness:** 3
**Presentation:** 4
**Contribution:** 3
**Rating:** 8
**Confidence:** 2

**Summary:**

This paper addresses the efficiency bottleneck of diffusion sampling by focusing on stepsize optimization, a relatively underexplored aspect compared to denoising direction refinement. The authors propose Optimal Stepsize Distillation (OSS), a dynamic programming (DP) framework that derives theoretically optimal stepsize schedules by minimizing global discretization error between a high-step “teacher” trajectory and a low-step “student” trajectory.

**Strengths:**

1. The paper formalizes stepsize optimization as a dynamic programming problem, which is theoretically elegant and clear.

2. The experimental design is comprehensive and solid, which includes both image tasks and video tasks.

**Weaknesses:**

No major concern, just suggestions or minor questions.

1. Evaluation metrics: The paper mainly reports PSNR and FID, without including perceptual or semantic alignment metrics (e.g., CLIP-Score, ImageReward, or human preference), which would reflect generation quality in other views.

2. Can the OSS framework be integrated jointly with solver optimizations such as DPM-Solver++ or UniPC in a unified search space, and if so, would further improvements emerge?


As I don't fully understand the theoretical analysis in this paper, I would rate 8 with a low confidence score.

**Questions:**

See Weakness.

---

### Official Review · Reviewer_3qCY · 2025-11-01

**Soundness:** 2
**Presentation:** 3
**Contribution:** 2
**Rating:** 2
**Confidence:** 3

**Summary:**

This paper proposes a dynamic programming idea for obtaining the optimal stepsize in diffusion sampling process via recursive approximation. Empirically, it demonstrates the method across multiple architectures and solvers. The idea leads to upto 10x accelerated Text to image generation without loss in quality.

**Strengths:**

- The idea of distilling a compact non-uniform schedule is pragmatic and useful.
- Empirical gains are promising and seem to have been tested on models including  Masked Autoregressive generation model and OpenSora.

**Weaknesses:**

It seems like this paper has been formatted poorly. It is a bit difficult to read. Missing details:
- L391 - The sentence is cutoff
- L393 - Visualization results for Open-Sora are promised but not available in the Appendix it looks like.
- Figure 5 does not seem to be referenced anywhere.


Other Weaknesses:
- Authors claim that the scheduler generalizes across ODE solvers but there is no ablation study to show this claim.
- PSNR evaluation is insufficient. Metric like FID needs to be also evaluated.
- Authors should also compare how much does the teacher trajectory generation + DP search cost compared to the sampling cost.
This factor may vary for various tasks and applications. Authors should consider discussing the costs of their method compared to plain sampling.

Overall, the motivation of this paper is good but the benefit is not clearly shown through experiments. Evaluations are not complete (missing metrics). Claims need to be supported with experiments and more evaluation metrics. This is why i am rejecting the paper.

**Questions:**

- Authors cite [Align Your Steps](https://arxiv.org/pdf/2404.14507) paper but do not compare with it. AYS does compare itself to deterministic solvers in Table-1. Can the authors comment why they chose to not compare?

---

### Meta-Review · Area_Chair_7JfM · 2026-01-05

**Summary:**

Two reviewers evaluated the paper as below the acceptance threshold (scores 2,4) and one reviewer evaluated it as above the threshold (score 8). The main concerns were: (1) lack of evaluations with perceptual metrics like FID (which is reported only for one experiment) and with prompt alignment metrics like CLIP-score; (2) lack of clarity regarding the computational burden and the time it takes to determine the optimal timestep schedule; (3) lack of comparisons to existing methods for step-size schedule optimization.

The authors did not upload a response.

The AC agrees with the points raised by the reviewers, especially regarding the missing discussion and comparison to previous work that targeted timestep schedule optimization. Furthermore, while the proposed approach seems effective, the AC finds that the presentation oversells its benefit. Claiming a $10\times$ acceleration for reducing from 100 steps to 10 steps is misleading because the 100-step baseline is arbitrary (one could also distill  $10^6$ steps into 10 steps and claim a $10^5\times$ acceleration, even though using $10^6$ steps probably does not provide much benefit over $20$ steps). The acceleration should be measured with respect to the number of steps that lead to the same quality as the distilled results. For Flux-dev, that is probably around 14 steps, which is only $1.4\times$ (and not $10\times$) slower than the paper’s 10-step results.

**Reviewer Concerns:**

No rebuttal was uploaded, so all reviewer concerns are outstanding.

**Reviewer Scores:**

All reviewer scores are expected to have remained the same because no rebuttal was submitted.

---

### Decision · Program_Chairs · 2026-01-26

Reject